# A Perspective of the Cumulative Risks from Climate Change on Mt. Everest: Findings from the 2019 Expedition

**DOI:** 10.3390/ijerph18041928

**Published:** 2021-02-17

**Authors:** Kimberley R. Miner, Paul Andrew Mayewski, Mary Hubbard, Kenny Broad, Heather Clifford, Imogen Napper, Ananta Gajurel, Corey Jaskolski, Wei Li, Mariusz Potocki, John Priscu

**Affiliations:** 1Climate Change Institute, University of Maine, Orono, ME 04463, USA; paul.mayewski@maine.edu (P.A.M.); heather.clifford@maine.edu (H.C.); mariusz.potocki@maine.edu (M.P.); 2Department of Earth Sciences, Montana State University, Bozeman, MT 59717, USA; mary.hubbard@montana.edu; 3National Geographic Society, Washington, DC 02917, USA; kbroad@rsmas.miami.edu (K.B.); imogen.napper@plymouth.ac.uk (I.N.); apgajurel@fulbrightmail.org (A.G.); 4Abess Center for Ecosystem Science and Policy, University of Miami, Coral Gables, FL 33146, USA; corey@synthetaic.com; 5Virtual Wonders, LLC, Wisconsin, Delafield, WI 53018, USA; 6School of Earth and Climate Sciences, University of Maine, Orono, ME 04463, USA; 7International Marine Litter Research Unit, University of Plymouth, Plymouth PL4 8AA, UK; 8Department of Land Resources and Environmental Sciences, Montana State University, Bozeman, MT 59717, USA; wei.li02@montana.edu (W.L.); jpriscu@montana.edu (J.P.)

**Keywords:** Mt. Everest, risk, climate change, pollution

## Abstract

In 2019, the National Geographic and Rolex Perpetual Planet Everest expedition successfully retrieved the greatest diversity of scientific data ever from the mountain. The confluence of geologic, hydrologic, chemical and microbial hazards emergent as climate change increases glacier melt is significant. We review the findings of increased opportunity for landslides, water pollution, human waste contamination and earthquake events. Further monitoring and policy are needed to ensure the safety of residents, future climbers, and trekkers in the Mt. Everest watershed.

## 1. Introduction

Mountain systems are changing rapidly throughout the world in response to climate change and human activity in general. Loss of water stored in glaciers, variations in timing and magnitude of precipitation, and warming at high altitudes impact alpine ecosystems and human inhabitants. High mountain glaciers worldwide are decreasing in extent and volume at dramatic rates with significant consequences for water availability, hazards (such as glacier lake outburst floods and slope failure, Figure 1), ecosystems, and socio-economic futures. Climate change in the Hindu Kush Himalaya is of particular concern given the region’s 250 million Nepali, Tibetan, and Chinese inhabitants, rich biodiversity, and some 2 billion more lives downstream who depend on the mountains’ bounty for water and food, among other services. With environmental changes occurring rapidly, we must elucidate potential risks for all people and ecosystems in the Hindu Kush Himalaya and mountainous regions worldwide.

Driven by climate change dynamics, known physical threats such as landslides, avalanches, and blizzards are compounded by hazards such as introducing distant and local source pollutants. These threats can impact a variety of environmental services critical to the stability of down-glacier populations. For example, both outburst flooding and the corollary or glacier wastage can affect water and water infrastructure availability and resilience. Landslides, avalanches, and blizzards present challenges to the physical safety of residents and supporting infrastructure. Polluted meltwater limits the available water for use in agriculture and by the resident populations.

At the confluence of these hazards, Mt. Everest and the Khumbu watershed present unique challenges from a significant inflow of tourists, trekkers, and local visitors. The health, safety, and livelihoods of residents and visitors alike are tied to this unique mountain system’s stability.

To illustrate first-order risks with the likelihood of increase under a warming climate, we present an overview of the hazards exacerbated by climate change on Mt Everest, where the ~10,000 residents and thousands of annual tourists, trekkers, and climbers witness and experience these changes first hand. The combination of inherent climbing risk and an increase in warming on the mountain introduces increased challenges to the Mt. Everest community. Unfortunately, research into global change dynamics is limited in mountainous regions, particularly above 5000 m, and the Himalaya contains many mountains above 8000 m.

National Geographic and Rolex’s Perpetual Planet Everest Expedition (hereafter the 2019 Everest Expedition) sought to complete novel, interdisciplinary research at and above 5,000m, including biology, geology, glaciology, mapping, and meteorology [1] to fill gaps in knowledge in these regions [2]. This 60-day expedition utilized diverse methodology to analyze water, soil, ice and general topography of the mountain [1,3,4]. While this methodology has been acknowledged in previously published work, the risk profiles of the diverse dynamics discovered have not been compiled. The majority of data presented in this study was collected by the 2019 Everest Expedition and represents a unique, yet still partial, snapshot of the mountain’s changes driven by climate change. Here, we use new data, combined with standing research, to highlight existing and emerging physical and chemical risks on Mt. Everest and the watershed below.

## 2. Physical Environment

### 2.1. Geologic Risk Formation

The spectacular high elevation mountains and topographic relief in the Himalaya are the product of the balance between crustal deformation in a convergent tectonic setting, and the impacts of erosional processes dissecting the Earth’s crust in this region [5,6]. Geological and hydro-meteorological (both the tectonic and erosional) processes become hazards in the Khumbu region as earthquakes, landslides, and flooding events [7]. As the Indian continent continues its generally northward movement with respect to the Asian continent, this interplay of processes shapes this mountain belt’s landscape. As a result, hillslopes tend to be steep, and rates of erosion are high, calibrated by the strength of the material being eroded, topography, vegetation, and extreme climatic conditions (Figure 1) [5,6].

In eastern Nepal’s Khumbu region, the rocks predominantly consist of gneiss, schist, quartzite, marble, and leucogranitic intrusions [8]. Given the history of faulting in the area, an abundance of precipitation, particularly during the monsoon, and freeze-thaw cycles, these rocks are prone to physical weathering through fracturing (e.g., ice wedging) [9]. The steep hillslopes are then susceptible to erosion via gravitationally driven events that range from slow and small-volume to fast and high-volume movements [6]. It is these high-volume, high-velocity landslides, including rockfalls, rockslides, and debris flows, that pose a geologic hazard to people and property in the Khumbu region [5,10,11]. Indeed, it has been documented that earthquakes often trigger landslides and avalanches of all sizes [12]. To understand the glacial and geomorphological changes critical to protecting both the climbers and villages of the Khumbu region, it is essential to understand the frequency and locations of past events. Toward this end, our team conducted a geochronological investigation of deposits that constrain the timing of landslide events near the villages of Namche Bazar and Phortse. By coring two lakes in the Gokyo valley, we also detected past hazard events from disturbances in the lake sediments. 

### 2.2. Landslide Risks

The Khumbu region landscape is shaped by glacial and fluvial erosion spanning from the last ice age to the present day. There is a diversity of landslide scarps and deposits, and the village of Namche Bazar (below Mt. Everest) was built in the bowl-shaped scarp from a past landslide [13]. That scarp cuts into an earlier massive rockslide--the Khumjung slide, which is currently the sixth-largest rockslide documented in Nepal [12]. This massive rockslide resulted from a 2.1 × 109 m^3^ volume of rock material sliding southward off the Khumbila peak, where remnants make up the small E–W-trending ridge that separates Khumjung and Kunde from Namche [4]. Rockslides of this magnitude are thought to account for a large percentage of the erosion across the high-relief Himalaya [13,14]. High-magnitude slides occur infrequently and may be triggered by earthquake events [14,15], where rock strength plays a role in the frequency of occurrence. Using the Infrared Stimulated Luminescence (IRSL) dating of feldspar grains from deposits that are cut by the slide, our team determined that the bowl-shaped slides of Namche Bazar happened before ~26 kyr. In contrast, the Khumjung slide occurred before ~26 kyr [16]. 

Rivers and Glaciers are erosional transport agents for rock material, breaking down bedrock outcrops and moving eroded material down the valley [17]. The Khumbu region typically has headwalls on the order of ~55°, the angle of which promotes large-scale rockfall events (catastrophic rock failure) in the highest parts of the range. As climate changes and glaciers retreat, they melt away from the headwalls, exposing steep rock faces and increasing their vulnerability to failure. A time series of remote sensing imagery from 1962 to 2019 reveals considerable ice loss in the Khumbu region, particularly the Imja and Baruntse valley glacier systems, with newly exposed rock risking catastrophic failure. All landslides, including those triggered by earthquakes and glacial recession, can introduce cascading hazards, including the damming of river valleys, rockslides, avalanches, and flood events [5,6,18,19].

### 2.3. Earthquake Risks

Rocks of the Khumbu region have been subjected to deformational processes for the past ~55 million years, creating regular failure planes. These planes can be due to a weaker lithology or prior faulting or shearing of the bedrock [20]. In the oldest examples of deformation in the Greater Himalayan Sequence, deep rocks were recrystallized at the mineral grain-scale during and after deformation [21]. More recent deformation episodes have developed planes of weakness through alignment of platy minerals or the fracturing or pulverizing of rock material. Once the incision of streams and glaciers steepens hillslopes, these weak planes are vulnerable to failure (Figure 2) [6,20]. The Khumbu region has several young faults that enhance the risk of a mass movement, including rock movement such as E-W striking thrust structures [18], E-W striking extensional structures [9,22], NE striking strike-slip structures [23,24].

Active faults at depth exacerbate landslide risk through earthquake occurrence [15,25]. Following the 2015 (Mw 7.8) Gorkha earthquake in Nepal, a debris avalanche consisting of ice, snow, and rock material buried the Langtang village, killing more than 350 people [12,26]. The earthquake caused a similar massive snow and ice avalanche at basecamp and killed more than 20 people. Kargel et al., (2016) mapped more than 4300 landslides triggered by that earthquake, including throughout the Khumbu and Dudh Kosi valley [10,12].

When coring lakes in the Gokyo valley, our team found multiple layers of disturbed sediment that may result from disturbance by earthquakes [27]. The 58 cm-long core covers ~1800 yrs suggesting that there have been numerous earthquakes large enough to disturb lake sediment during that period. This frequency of earthquakes is consistent with other evidence for earthquakes found throughout the Himalaya and suggests that earthquakes will continue at this regularity [5]. Earthquakes compounded with warming can increase the opportunity for landslides and outburst floods.

There are too many variables in our Earth’s heterogeneous crust to be able to predict earthquakes precisely. Still, seismologists have done significant work studying past earthquakes and rates of modern tectonic plate movement. This research allows scientists to understand areas along the Himalaya overdue for slip events and at higher risk of earthquakes [22,28]. Like all other parts of the Himalaya, the Khumbu is susceptible to future high-magnitude earthquakes and the significant risks to humans and wildlife those events could cause. Seismic events may increase the probability of mass movement of both rock and ice, highlighting the interconnectivity of high mountain hazards in tectonically active mountain belts.

### 2.4. Glacial Lake Outburst Flooding Risks

Glacial lake outburst floods are another geologic hazard that relates to the evolving geomorphology under a warming climate. Outburst floods occur as glaciers retreat and leave newly formed lakes impounded by their former moraine walls. These lakes risk catastrophic breakthrough of this impoundment and subsequent flooding. Outburst floods of glacial lakes are most often triggered by mass movements into the lakes [29], including avalanches, landslides, or rockfall (Figure 1) [30,31]. Evidence from sediment coring of proglacial Lake Gokyo demonstrates that mass movements into lakes have occurred, even at high elevations (4750 m) [32]. As glacial melt accelerates, the formation of supraglacial lakes on the glacier surface becomes a hazard [33] as these lakes are isolated from meltwater release points. If supraglacial lakes suddenly break their banks, they can release significant water flow and downstream flooding [19].

The Khumbu region of Nepal, including Sagarmatha National Park, is host to several geologic hazards and contains a cryptic record of past events. This mountain region includes the highest elevations globally, and it balances tectonic uplift with erosional processes. The result is a network of steep glacial headwalls with active erosion and stream incision. As our climate changes, we see the retreat of these glaciers, growth of glacial lakes, and heightened risk from mass movements and flooding events. The seismic hazard is ever-present throughout the Himalaya and heightens the risk of mass movements and avalanches. The highest risk areas are often concentrated along the river valleys or steep valley walls, which host most of the settlements in the Mt. Everest watershed and beyond, enhancing the potential for human impacts.

### 2.5. Physical Hydrology Risks

Superimposed on the geological risks of landslides, earthquakes, and glacial lake outburst floods, hydrological conditions can also become hazardous in a changing climate. However, high-elevation continuous hydrological monitoring is scarce. Our team installed a network of five automatic weather stations (AWSs) to investigate water resources and temperature in the context of climate change and to improve climber safety [34]. Observations indicate exceptionally high values of incoming solar radiation that can cause surface melting at over 8000 m even with ambient air temperatures well below freezing where melting and high rainfall may lead to landslides and flooding [34]. This widespread melt may mobilize PFAS, microplastics, and other hazardous substances in the Everest ecosystem (Figure 3) [35,36].

## 3. Chemical Risks

### 3.1. Legacy Chemical Pollution Risks

The global signature of chemical pollution emergent from the cryosphere is increasing as melt progresses [35]. High alpine and Polar environments see the most disproportionate warming [37,38], but are also the most critical global water towers, distributing water to millions of people [39].

On Mt. Everest, our research has uncovered the signature of generations of trekkers, in the form of directly deposited per-fluoroalkyl substances, or PFAS, compounds used to coat plastics for waterproofing [3]. These compounds have been used widely since the 1950s and are ubiquitous in outdoor gear as varied as coats, boots, and tents (Figure 1).

However, the deposition of PFAS into the Everest ecosystem has left a concentration levels two orders of magnitude above background levels across the mountain landscape [3]. Indeed, at the levels detected, human consumption of the PFAS in Basecamp are at levels just below regulated concentrations in the United States and European Union [40]. All of the meltwater utilized by climbers and guides at basecamp is collected from the local ecosystem and only treated for microbes, leading to direct uptake of PFAS in the watershed.

PFAS are only one compound found in quantity across the mountain. Our research has also uncovered DDT, Lindane, and lead. All of these compounds were found in amounts just below the screening levels for health impacts [41,42]. However, in combination, they could pose an increased risk to trekkers and the local community. While health impacts from one-year of water consumption may be minimal, the regular uptake of pollutants by Sherpas and guides could increase risk significantly through bioaccumulation [43,44]. In future research, sampling for toxic chemicals and metals must be prioritized, with researchers likely to find many more compounds left by humans across the Khumbu region’s landscape.

### 3.2. Heavy Metal Pollution Risks

Heavy metals (e.g., Pb, As, Cd, Cr) occur naturally in the environment, yet anthropogenic activities may increase levels, with potential adverse effects on human health, biodiversity, and ecosystems. Some heavy metals are classified as systemic toxins and can cause adverse health effects, including organ damage, cardiovascular disease, developmental abnormalities, cancer, and neurological disorders. Macro-scale human activities significantly amplify long-range and local transport of atmospheric metals (e.g., fossil fuel combustion, metal production, waste production) and local activities (e.g., agriculture, expansion in land use, biomass burning). Heavy metals, attributed to long-range transport of aerosols from human sources [45,46], have been previously reported in Himalayan studies of snow and ice [47,48,49,50,51,52,53]. Ice core records from the north side of Everest demonstrate that some heavy metals have increased since the 1950s due to anthropogenic activities [53,54].

Heavy metals accumulate with dry and wet deposition on the surface of glaciers, decreasing surface albedo and resulting in glacier outburst floods and glacier volume loss. Entrapped metals are released as snow and glaciers melt, contaminating local watersheds, affecting ecosystems and agriculture [39] if metals exceed safety levels. Climate change can also contribute to increased metals in watersheds by increased weathering, melting permafrost, and falling water tables [55,56]. Any increase in downstream concentrations in heavy metals may impact the local population, as glacier melt from the Khumbu glacier accounts for an average of 65% of water resources during the dry, pre-monsoon season [57].

During the 2019 Everest Expedition, water and snow sampling revealed the chemical footprint of anthropogenic metals from both local and long-range sources. Elevated pollution-based metal and other aerosol concentrations in surface snow, with specific increases near local villages, compared to more remote sample sites. Concentrations of specific metals in the local glacial stream system, sourced from the Khumbu glacier, indicate sufficient evidence to necessitate future monitoring. Among the toxic metals detected, several snow and stream samples contained concentrations of metals above WHO drinking water safety level guidelines. A recent study found concerning heavy metals levels exceeding WHO guidelines in local Gokyo lake water samples [58], while other studies in the region have not detected heavy metal contamination in surface waters [59,60,61].

### 3.3. Pathogens Pollution Risks

Pathogens transported by glacial melt through the below glacier watershed may be an area of increased importance as the climate warms. Castello and Rogers (2005) estimated an annual release of 1017–1021 viable microorganisms globally from glacier melt into the downstream environments from atmospheric deposition and in-situ growth [62,63,64,65,66]. Atmospheric aerosols are comprised of a wide range of biological materials, including Bacteria, Archaea, pollen, fungi, protists, and viruses [67,68], some of which are plant or human pathogens (Figure 1) [69]. The aerosols and associated pathogens are transported and deposited via precipitation onto glaciers far from origin [70,71,72]. As glaciers and ice sheets recede at an accelerated pace in response to climate change, there are concerns about the potential impact of the enhanced release of ice-entombed pathogenic organisms on downstream human populations [62]. The release of pathogenic bacteria has been noted by reports of glacial transport of fecal bacteria from climbing activities on Mt. McKinley’s Kahiltna Glacier, Alaska. In this case, buried human waste from climbers emerged at the glacier surface within decades of deposition [73,74].

We analyzed environmental DNA [75] in ice cores from Mount Everest South Col (ESC, at 8030 m asl) and Mount Everest Base Camp (EBC, at 5400 m asl) to determine the presence of potential pathogens along this popular climbing route. Bacterial operational taxonomic units (OTUs) closely related to Clostridioides difficile (98.5% similarity, NCBI accession NR_113132) and Pseudomonas aeruginosa (99.2% similarity, NCBI accession NR_117678) were identified in sequence libraries derived from ice core samples collected during the expedition. Both species have the potential to impact human health. Clostridioides difficile is a spore-forming toxin-producing Gram-positive anaerobic bacillus that can cause toxin-mediated infections ranging from mild diarrhea to death dominant transmission through ingestion of spores [76]. Pseudomonas aeruginosa is a Gram-negative bacterium found in various habitats, including soil, marshes, marines, plant, and animal tissues, and is noted for its resistance to antibiotics and disinfectants. Owing to this resistance that allows for decreased competition, P. aeruginosa is a primary opportunistic human pathogen known to threaten burn victims, urinary-tract infections patients, and pneumonia patients [77].

OTUs closely related to Streptococcus vestibularis (99.2% similarity, NCBI accession NR_042777) were identified only in the highest ice core (8030 m). S. vestibularis is a Gram-positive coccus, forming a significant part of the healthy human oral cavity, but is known in association with infective endocarditis [78]. Sequences closely related to Staphylococcus epidermidis (99.2% similarity, NCBI accession NR_113957) were found only in the Base Camp ice core sample. S. epidermidis is a Gram-positive opportunistic human pathogen that often infects patients with compromised immune systems [79].

Mitochondrial 16S ribosomal-RNA genes related to two pathogenic fungi were also identified, including Fusarium oxysporum (NCBI accession KU158767), a fungal pathogen to numerous agricultural plants [80] and Paecilomyces penicillatus (MK069583), which infects other fungal species [81]. Although the source of these pathogenic strains within the ice core samples analyzed remains unclear, results indicate that glacial meltwater consumption may lead to illness in local and downstream human, animal wildlife, and plant populations. Further research and indexing of pathogenic species are necessary as the biological waste left by generations of climbers on the glacier continues to move downstream.

### 3.4. Plastic Pollution Risks

Nepal’s tourism business and livelihood diversification have supported the local economy in the Khumbu region [60,82,83]. However, with this intensity of tourism, the potential for conflict between maintaining a healthy natural environment and development also increases [82]. The first climbers to Mt. Everest began a century ago, ten years after the South Col (~8000 m) on Mt. Everest was considered the highest junkyard in the world [83]. More recently, the field team from this expedition found many items such as plastic bottles, oxygen bottles, food wrappers, food waste, and cigarette butts [4].

In recent decades, innovation and application of new plastics led to the introduction of lightweight, technical clothing and equipment, making the mountain more accessible [84]. Unfortunately, to date, a large proportion of the accumulating waste is plastic due to these lightweight, strong, inexpensive, durable, and corrosion-resistant properties.

Our research has identified that the increase in climbers over the last 50 years has resulted in associated increased accumulation of anthropogenic waste, including litter along trekking trails and microplastic build-up [4,36]. Our teams found microplastics throughout samples on the mountain, suggesting that deposition may be ubiquitous [4].

Given the fragile environment, extreme weather conditions, and limited infrastructure, the waste produced is often beyond local capacity to handle, increasing local towns and economies’ stress [82]. New waste management systems are required, as the complete elimination of waste locally would be extremely challenging. Plastic pollution joins the growing chemical load on the mountain as an additional challenge to local populations unable to manage the challenges associated with generations’ influx of trekkers.

However, there have recently been a series of positive actions to address risks from waste build-up. For example, in February 2019, China closed its Everest base camp on the Qinghai-Tibetan Plateau due to waste accumulation. Starting in January of 2020, the government banned single-use plastics on the mountain to reduce waste left by climbers. According to recent legislation, the Government of Nepal also introduced a provision for garbage management in its Mountaineering Expedition Rules, 2002 that the deposit of $4000 will be returned only after the submission of evidence of garbage management when the mountaineers descended from Mount Everest.

With the expansion of tourism and infrastructure development, local emissions and long-range transport could raise contaminant concentrations in surface snow and glacier melt streams in the Khumbu region. For long term management, increased spatiotemporal environmental monitoring throughout the below glacier watershed is recommended, and expanding local regulations on tourism, biomass emissions, and waste removal.

## 4. Conclusions

For all of the people and ecosystems in the Himalaya, it is critically important to clarify potential risks and to incorporate these into mitigation and adaptation planning. The impacts left by humans expressed through climate change, land-use changes, and the degree of human occupation across landscapes, even on the highest mountain on Earth, demonstrate the far-reaching consequences of our activities. This human signature becomes increasingly harmful as glaciers melt at an unprecedented rate throughout the world. The 2019 Everest Expedition adds to existing high mountain research by providing a framework for combined research in biology, geology, glaciology, mapping, and meteorology and a demonstration of the necessity to expand our understanding of the diversity of risks that mountain regions pose and the need to monitor the status of these changing conditions.

The growing ecosystem changes expose humans’ hidden impacts across landscapes, where our influences are seen even on the highest mountain on Earth. This frozen chemical signature of human interaction is an unfortunate side effect of innovation that becomes increasingly harmful as glaciers melt at an unprecedented rate. In the future, it will be necessary to continue to expand our understanding of the diversity of risks that the mountain poses and to monitor the status of these changing conditions.

The compounding physical and chemical risks discovered on the mountain have the potential to magnify over the next 10 years. Increased glacial melt creating geologic instability and chemical pulses may drive system change as the hazard exposure increases to a greater area. Already, the number of risks that climbers and residents face are increasing with the warming climate, and exponential yearly change is possible. Future research must prioritize the immediate risks to the health and livelihood of residents, followed by the climbing and trekking community.

Though the risks to health and safety have always been a part of climbing Mt. Everest, the changing dynamics implicit within a warming climate compound these dynamics and increase the hazards present. Both governmental and non-government organizations must understand the regional impacts of climate change to mitigate the effects of mountaineering and climate warming successfully. Solving this time-sensitive problem is an essential and critical and understanding the sustainable way forward for the greater Himalaya community.

## Figures and Tables

**Figure 1 ijerph-18-01928-f001:**
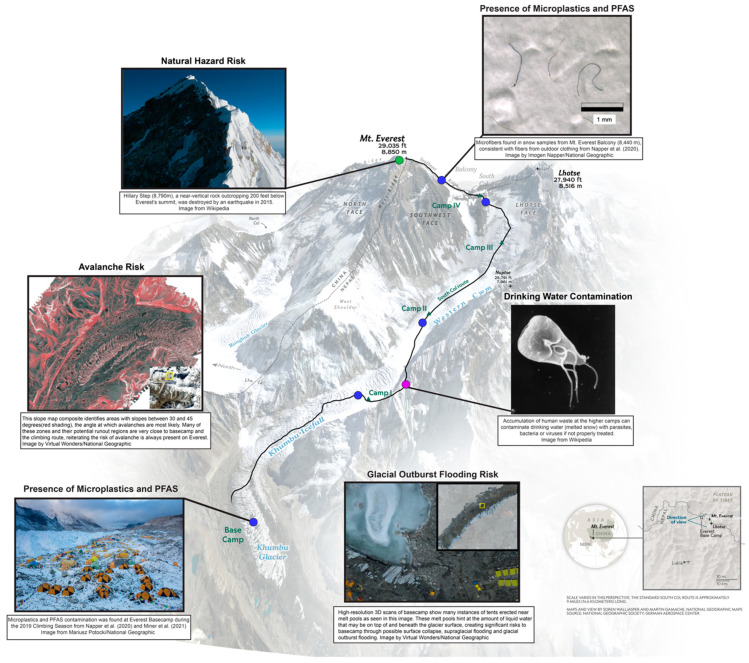
Map of Mt. Everest ascent with representative risks highlighted. Blue dots indicate the presence of Perfluorooctanoic acid (PFAS) or microplastics, red dots indicate hydrological risks, green dots indicate geological hazards and purple dots indicate human physiological water contamination. The locations are representative and not exclusive to the hazard represented.

**Figure 2 ijerph-18-01928-f002:**
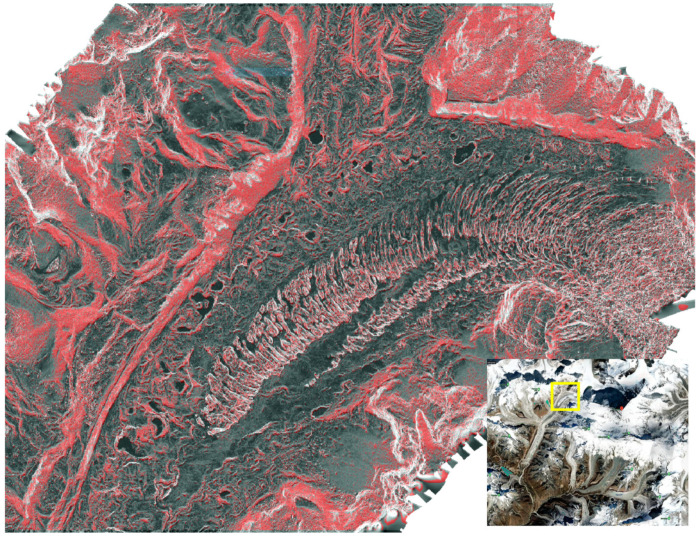
Close-up of slope composite map taken by helicopter at Khumbu glacier fall. This red highlighting indicates a location has significant (35-40 degree) incline, but is representative of multiple slopes up glacier. Image created by Virtual Wonders and National Geographic.

**Figure 3 ijerph-18-01928-f003:**
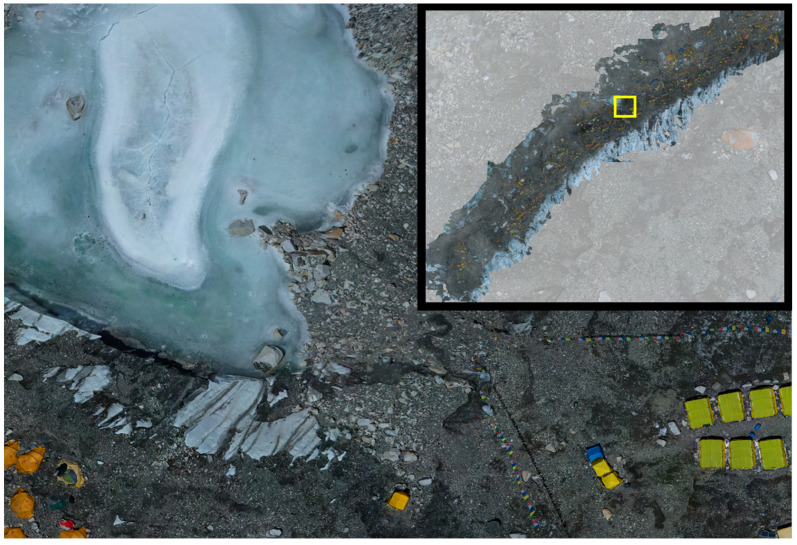
Close-up aerial view of basecamp showing where tents are erected. Darker soil indicates supraglacial water flow. Image by Virtual Wonders and National Geographic.

## Data Availability

All data is available through the National Geographic Society and cited publications.

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
