# Peer review of "A Perspective of the Cumulative Risks from Climate Change on Mt. Everest: Findings from the 2019 Expedition"

_ijerph, 2021, doi:10.3390/ijerph18041928_

Round 1

Reviewer 1 Report

1.The paper should briefly introduce the basic situation of the 2019 Mt. Everest expedition, such as how many days of field work, how to scientifically plan the route of the expedition, what kind of technical methods are used, and from what aspects scientific data are collected.

2.The means of the field investigation of the Mt. Everest expedition are more important. This paper studies a very macro perspective. Along the planned artificial investigation route, that is, using UAV to collect data, the scope of vision is relatively limited. Further, I think that large-scale risk / source identification can be analyzed by combining with remote sensing data.

3.The section 2.1 seems the contents of section 2.2-2.5, which is not parallel with section 2.2-2.5, but an overview? Then it is suggested to cancel the section number.

4.By reference to “3. Chemical risks”, is better named "2. Physical risks" than “2. Physical environment”?

5.Line 388 in page 8,”3.2. Figures, Tables and Schemes”,The section number is incomprehensible.

6.It is suggested to add a discussion part. Including: 1) What are the cumulative relationship between the cumulative risks from climate change? 2) Combined with previous research results, this paper further analyzes the change trend of these risks in recent ten years, which will help to provide scientific basis for risk control decision-making.

Author Response

Thank you for your helpful comments, please see responses below.

  1. This section has been adjusted to accommodate information on methodology and previous publications

"National Geographic and Rolex’s Perpetual Planet Mt. Everest Expedition (hereafter the 2019 Everest Expedition) sought to complete novel, interdisciplinary research at and above 5,000m, including biology, geology, glaciology, mapping, and meteorology1 to fill gaps in knowledge in these regions.2 This 60-day expedition utilized diverse methodology to analyze water, soil, ice, and general topography of the mountain.1,2,38,82 While this methodology has been acknowledged in previously published work, the risk profiles of the diverse dynamics discovered have not been compiled. The majority of data presented in this study was collected by the 2019 Everest Expedition and represents a unique, yet still partial, a snapshot of the mountain's changes driven by climate change. Here, we use new data, combined with standing research, to highlight existing and emerging physical and chemical risks on Mt. Everest and the watershed below."

2. Thank you for this comment. While the expedition did not include remote sensing at this time, it is a very useful suggestion for future work.

3. Thank you, it appears the sections numbers have been made uniform by the MDPI team

4. Thank you, we have changed the section title to reflect your suggestion

5. Thank you, the MDPI team has changed the numbering to make it more clear

6. Thank you for this comment. Though this paper has no discussion section, we have added the following to the conclusions to incorporate your ideas.

The compounding physical and chemical risks discovered on the mountain have the potential to magnify over the next 10 years. Increased glacial melt creating geologic instability and chemical pulses may drive system change as the hazard exposure increases to a greater area. Already, the number of risks that climbers and residents face are increasing with the warming climate, and exponential yearly change is possible. Future research must prioritize the immediate risks to the health and livelihood of residents, followed by the climbing and trekking community. Though the risks to health and safety have always been a part of climbing Mt. Everest, the changing dynamics implicit within a warming climate compound these dynamics and increase the hazards present. Both governmental and non-government organizations must understand the regional impacts of climate change to mitigate the effects of mountaineering and climate warming successfully. 

Reviewer 2 Report

This is a useful descriptive presentation of some of the preliminary results obtained from a one season expedition to Mount Everest. As it is quite hard to get data from high altitude sites, such additional data are always welcome. Given the very short description of the methods used (for example how different cores used are obtained, both for sediments and ice, sampling methods to assess chemical contaminants, etc), it is hard to assess the robustness of the results, but they will probably be presented in more specialized journals. I would have appreciated to see a bit more data than what was available in the current version (eg on actual temperatures, radiation, chemical concentrations, ...).

You might consider changing the order in the abstract when writing "ensure the safety of future climbers, trekkers and residents in the Mt. Everest watershed." - perhaps residents should come first. As you make clear for example regarding chemical contaminations, they may represent a more important risk for residents (even if sherpas are also climbers, of course). And I would argue that residents come before trekkers.

In my version of the paper, there was a whole section after l. 373 that was a copy and paste from the instruction to authors.

Author Response

Thank you for your review and comments.

  1. We have added additional information on the types of samples retrieved with citations to the original papers and motivations for this paper:

"

National Geographic and Rolex’s Perpetual Planet Mt. Everest Expedition (hereafter the 2019 Everest Expedition) sought to complete novel, interdisciplinary research at and above 5,000m, including biology, geology, glaciology, mapping, and meteorology1 to fill gaps in knowledge in these regions.2 This 60-day expedition utilized diverse methodology to analyze water, soil, ice and general topography of the mountain.1,2,38,82 While this methodology has been acknowledged in previously published work, the risk profiles of the diverse dynamics discovered have not been compiled. The majority of data presented in this study was collected by the 2019 Everest Expedition and represents a unique, yet still partial, snapshot of the mountain's changes driven by climate change. Here, we use new data, combined with standing research, to highlight existing and emerging physical and chemical risks on Mt. Everest and the watershed below."

2. Absolutely, this is very valid and we have made the appropriate change

"In 2019 the National Geographic-Rolex Mt. Everest expedition successfully retrieved the greatest diversity of scientific data ever from the mountain. The confluence of geologic, hydrologic, chemical and microbial hazards emergent as climate change increases glacier melt is significant. We review the findings of increased opportunity for landslides, water pollution, human waste contamination and earthquake events. Further monitoring and policy are needed to ensure the safety of residents, future climbers, and trekkers in the Mt. Everest watershed."

3. Apologies, this was due to a software glitch and has been remedied